# Interaction Between Neutrophils and Elements of the Blood–Brain Barrier in the Context of Multiple Sclerosis and Ischemic Stroke

**DOI:** 10.3390/ijms26094437

**Published:** 2025-05-07

**Authors:** Anna Nowaczewska-Kuchta, Dominika Ksiazek-Winiarek, Andrzej Glabinski

**Affiliations:** Department of Neurology and Stroke, Medical University of Lodz, ul. Zeromskiego 113, 90-549 Lodz, Poland; anna.nowaczewska-kuchta@umed.lodz.pl (A.N.-K.); dominika.ksiazek@umed.lodz.pl (D.K.-W.)

**Keywords:** neutrophils, endothelial cells, astrocytes, pericytes, multiple sclerosis, ischemic stroke

## Abstract

The blood–brain barrier (BBB) is a semi-permeable membrane in physiological conditions, but in pathologies like multiple sclerosis (MS) and ischemic stroke (IS), its permeability increases. In this review, we focus on neutrophils and their interaction with cellular components of the BBB: endothelial cells (EC), pericytes (PC), and astrocytes (AC). Nowadays, neutrophils receive more attention, mostly due to advanced research techniques that show the complexity of their population. Additionally, neutrophils have the ability to secrete extracellular vesicles (EVs), reactive oxygen species (ROS) and cytokines, which both destroy and restore the BBB. Astrocytes, PCs, and ECs also have dual roles in the pathogenesis of MS and IS. The interaction between neutrophils and cellular components of the BBB provides us with a wider insight into the pathogenesis of common diseases in the central nervous system. Further, we comprehensively review knowledge about the influence of neutrophils on the BBB in the context of MS and IS. Moreover, we describe new therapeutic strategies for patients with MS and IS like cell-based therapies and therapies that use the neutrophil function.

## 1. Introduction

The blood–brain barrier (BBB) is a membrane between the tissue of the brain and the blood. This membrane is selectively semi-permeable and regulates movements of ions and other factors [1]. The BBB is created by microvascular endothelial cells lining the cerebral capillaries penetrating the spinal cord and the brain [2]. The BBB comprises endothelial cells (ECs), astrocyte end-feet, pericytes (PCs), and capillary basement membrane [1].

Endothelial cells are one of the components of the BBB. These cells create a surface between circulating blood or lymph in the lumen and the rest of the vessel wall. In basal conditions, endothelial cells participate in thrombosis and thrombolysis, platelet adherence, angiogenesis, and the regulation of inflammatory responses by modulating the interaction between leukocytes and the vessel wall [3]. They secrete growth, vasomotor, antithrombotic, inflammatory, and procoagulant factors and matrix products [3,4]. Loss of proper endothelial function plays a great role in the pathogenesis of atherosclerosis [5]. Endothelial dysfunction is characterized by reduced vasodilatation, the presence of proinflammatory state, and prothrombotic properties. These changes are associated with cardiovascular diseases such as stroke, hypertension, diabetes and other disorders [6]. In addition, endothelial dysfunction leads to vascular inflammation, which contributes to the development of multiple sclerosis (MS) by BBB disturbances and increased leukocyte migration [7].

Astrocyte (AC) is the next component of the BBB. Astrocytes are the key class of glial cells, which are star-shaped. They are divided into subtypes: fibrous, protoplasmic, and radial [8]. Astrocytes regulate vascular tone and local blood flow into the brain parenchyma, and hence play an important role in the transport of oxygen and nutrients to neurons to maintain brain homeostasis [9]. These cells also maintain ionic homeostasis of the extracellular space, regulate pH and neurotransmitter uptake, and mediate signals from neurons to the vasculature [10]. Changes in AC characteristics contribute to the impairment of BBB integrity [11]. A study conducted on mice showed that astrocytic expression of the vascular endothelial growth factor (VEGF) was a key factor of BBB permeability [12]. In pathologic conditions, astrocytes transverse in reactive astrocytes. Changes in their reactivity, gene expression and functional characteristics have been observed. After stroke and in MS lesions, astrocytes may aggravate neuroinflammation but also promote neurological recovery and lesion repair [13,14]. Neurodegenerative diseases such as Alzheimer’s disease (AD) and Parkinson’s disease (PD) are also associated with pathologies of astrocytes [15].

Pericytes are the next part of BBB. A PC cell body contains several long processes surrounding the abluminal endothelial wall [16]. The essential function of PCs is to maintain the integrity of vessels and the BBB. It was confirmed that in hypoxia, they protect the function of BBB and keep it stable [17]. PCs participate in angiogenesis, the constriction and dilation of capillaries, the regulation of immune cell migration, and support the survival of ECs [18,19]. PCs lose proper functions in many diseases like stroke, depression, epilepsy, diabetes, AD, MS, and glioma [20]. In an early stage of human MS, loss of BBB integrity is associated with PC degeneration [21].

The main function of the BBB is the formation and maintenance of homeostasis for neurons. Moreover, the BBB defends the central nervous system (CNS) against toxic molecules, providing nutrition for the brain and communication between the CNS and the periphery [1]. Under the right physiological conditions, the BBB is relatively impermeable [2]. In pathological conditions, a BBB disruption is observed after ischemic infarction or brain injuries. BBB dysfunction is associated with changes in interaction between components of the BBB in active MS lesions [22]. Alterations in the BBB have been noted in neurodegenerative diseases such as AD or PD, but also in diseases like diabetes, obesity and, hypertension [23,24,25].

Neuroinflammation leads to the activation of ECs in the brain, which secrete inflammatory cytokines and have ability to antigen presentation [26,27,28]. Moreover, it decreased the expression of junction proteins [29]. Also, ACs regulate the BBB function during neuroinflammatory reaction by secreting pro-inflammatory cytokines, and their end-feet can be detached from the BBB by the influence of MMPs [30,31,32]. Moreover, PCs release inflammatory factors and have a phagocytic ability [26]. Overall, neuroinflammation through these processes increases BBB permeability, resulting in the infiltration of peripheral cells into the CNS, and impairs axons and myelin structure and function [26,27,28].

Multiple sclerosis is an autoimmune, demyelinating disease in which the myelin sheaths of nerve cells in the brain and spinal cord are destroyed [33]. There are various clinical phenotypes of MS: clinically isolated syndrome, primary progressive MS, secondary progressive MS and relapsing-remitting MS [33]. In the pathogenesis of MS, disruption of the BBB is a initial critical step [9]. BBB breakdown and transendothelial migration of activated leukocytes are observed in the early stage of MS lesions [34]. The BBB disruption is thought to be transient but recurrent at the same or different locations over time, causing MS lesions [34]. Even though the pathogenesis of MS is still not fully recognized, neutrophils gain more attention among researchers, as studies have shown an increased number of neutrophils in the periphery and in cerebrospinal fluid before and during the onset of the experimental autoimmune encephalomyelitis (EAE) model [35].

Ischemic stroke (IS) is defined by the occlusion of blood vessels in the brain, and this disease is the third most common cause of global disability and the second most common cause of global deaths [36]. Ischemic stroke occurs twice as often as hemorrhagic stroke worldwide [36]. Neutrophils are one of the first cells to appear in the CNS after stroke. Factors like reactive oxygen species (ROS), myeloperoxidase (MPO), and protease, produced by neutrophils, induce ECs damage, which results in increased permeability of the BBB [37]. Moreover, the effect of hypoxia–ischemia, which occurs in ischemic stroke, causes the disruption of the tight junctions (TJs) and escalates BBB permeability [38]. The breakdown of the BBB leads to the migration of cytotoxic substances and may cause brain oedema and hemorrhagic transformation [39].

The aim of our article is to analyze the influence of the neutrophils on the cellular components of the BBB. Moreover, we show the impact of endothelial cells, astrocytes and pericytes on pathogenesis of MS and IS. There are many neurodegenerative diseases that are connected with BBB disruption. However, we have chosen these two brain diseases to discuss two various pathomechanisms. Firstly, multiple sclerosis as autoimmune disease, and secondly, ischemic stroke as vascular disease with hypoxia–ischemia mechanism.

## 2. Neuroinflammation-Induced BBB Dysfunction in MS and IS

Neuroinflammation is an inflammatory process in the CNS, which plays a crucial role in the pathogenesis of both multiple sclerosis and ischemic stroke. Residual CNS cells, cellular elements of the BBB and peripheral cells, which cross the BBB, but also other factors, like cytokines, chemokines, ROS, and MMPs, are involved in this process [40,41,42,43,44,45]. During neuroinflammation in CNS disorders, astrocytes recognize and integrate signals of neuronal injury to regulate neuroinflammatory reaction. Astrocytes have the ability to create physical and molecular barriers, which seal and localize the damage area [32]. Pericytes in the neuroinflammatory milieu release inflammatory factors like MMPs, ROS, chemokines and cytokines, which leads to BBB disruption. Moreover, PCs are mediators of neuroinflammation due to their phagocytic ability and antigen presentation [26]. During neuroinflammation, ECs demonstrate the ability present antigens, release chemokines and cytokines, and also influence the permeability of the BBB [26].

MS is an autoimmune, demyelinating disease, in which neuroinflammation results from dysfunction of the immune system. The pathogenesis of MS remains unknown, but genetic, pathogenic and environmental factors, such as specific viral infection and other pathogens, may be involved. These factors may induce systemic inflammatory reaction [27,28,34]. This process leads to demyelination, in which myelin sheaths of nerve cells in the CNS are destroyed [33]. The immune process activates adhesion molecules of endothelial cells in the brain vascular system, which, in turn, leads to the infiltration of peripheral cells like lymphocytes, neutrophils, and macrophages via the BBB and impairs myelin and axons [27,28]. Neutrophils are one of the first cells that migrate into the CNS from the periphery, both in MS and IS [37,46,47]. Pericytes enhance leukocyte recruitment through the BBB and increase BBB permeability by the secretion of factors such as CXCL-8, IL-8, ROS, and MMPs [48,49,50]. Additionally, astrocytes, by secreting pro-inflammatory cytokines, such as IL-1β, IL-1α, and TNF-α, and by promoting inflammatory milieu astrocytes, facilitate BBB disruption and cause the infiltration of T- and B-lymphocytes, macrophages, and neutrophils into the CNS [30,42]. Moreover, astrocytes participate in inducing the alteration of CD4+ and CD8+ T cells into pro-inflammatory types characterized by cytotoxic activity [51]. Both CD4+ and CD8+ T cells are involved in MS pathogenesis [52].

In the acute stage of IS, neuroinflammatory reaction, which is associated with oxidative stress, influences neuronal injury [53]. The inflammatory mechanism begins after vessel occlusion via an interaction between P-selectin in endothelial cells and in α-granules in platelets [54]. Then, the activated coagulation cascade leads to the attraction of neutrophils and monocytes. This intravascular inflammation initiates BBB disruption and leukocyte infiltration to the ischemic area [29]. Also, the expression of the factors junctional adhesion molecule-A (JAM-A) and JAM-B by pericytes enables neutrophils to migrate into the CNS [29]. Furthermore, pericytes release MMP-9, which causes increasing permeability of the BBB [55]. After ischemia, astrocytes are activated by cytokines derived from neurons, and they secrete IL-1β, CCL2, and MMP-2, which leads to BBB damage and the formation of glial scars [56].

## 3. Neutrophils

Neutrophils are the most numerous type of white blood cells [57]. They are called polymorphonuclear (PMN) leukocytes, and their main function is to defend the body against invading factors [58]. In order to eliminate pathogens, neutrophils use phagocytosis, degranulate the contents of their granules, and produce neutrophil extracellular traps (NETs) [30,58]. Until recently, this function was considered the only goal of neutrophils. Nowadays, it is known that neutrophils are transcriptionally active complex cells, modulating the activities of other cells and contributing to tissue repair and the resolution of inflammation [59,60]. Additionally, they actively attend plenty of diseases, including cancer, MS and stroke [61,62]. They are one of the first cells to appear in the brain during inflammation. The production of NETs and the secretion of inflammatory factors like ROS, lipocalin 2 (LCN-2), MPO and matrix metalloproteinases (MMPs), especially MMP-9 by neutrophils, reduce tight junction proteins (ZO-1, occludin, claudin-5) and BBB breakdown [30]. Interestingly, after apoptosis, neutrophils release damage-associated molecular patterns (DAMP), which leads to cascades of Nod-like receptor-containing pyrin domain 3 (NLRP3)-inflammasome activation in other cells, causing a cycle of neurotoxicity and cell death. In consequence, this process leads to increased maturation of proinflammatory factors (IL-1β and IL-18) and exacerbates chronic CNS diseases [63]. Additionally, neutrophils, as most of other cells, are able to secrete extracellular vesicles (EVs) into extracellular space [64]. The main subtypes of EVs are exosomes, microvesicles, and apoptotic bodies. Their sizes and contents are different, and they perform different functions [65]. EVs derived from neutrophils are reported to show pro- and anti-inflammatory properties, depending on the types of stimuli and neutrophils, and can easily cross the BBB [64,65]. Moreover, changes in neutrophil phenotypes were observed during tumor progression, and on the basis of these results, various subtypes of these cells were called tumor-associated neutrophils (TANs). The TAN distinguishes two groups: N1 phenotype and N2 phenotype. This classification was later implemented into physiological and pathological states [66,67]. Tumor-suppressing N1 neutrophils release pro-inflammatory factors like tumor necrosis factor-α (TNF-α), IL-12, CXCL9, and CXCL10 [68,69]. N2, contrary to the N1 phenotype, demonstrates tumor-promoting and immunosuppressive activity. N2 neutrophils act via oncostatin M, MMPs, neutrophil elastase (NE) [69]. Other neutrophils have the ability to return to the bloodstream from the tissue in the process, which is called reverse transendothelial migration (rTEM) [70]. Neutrophils rTEM have prolonged life-span and pro-inflammatory features. They produce an increased number of NETs and may change a local inflammation into a systemic inflammation [71,72].

## 4. Neutrophils and Endothelial Cells

Brain microvascular endothelial cells present distinctive morphological, structural, and functional characteristics that distinguish them from other vascular endothelial cells [73]. These include the expression of tight junction proteins (sealing the paracellular pathways between adjacent endothelial cells), the absence of fenestrations and lack of pinocytic activity as well as expression of active transport mechanisms allowing the passage of essential molecules (including nutrients and essential amino acids), and blocking the passage of potentially undesired substances (both endogenous and xenobiotics) [73]. A distinguishing feature of ECs is the fact that they possess relatively few caveolar vesicles, which limit transcellular transport [9]. These cells are also considered to form an endocrine secretory tissue that secretes a variety of local hormones, such as nitric oxide, prostaglandins, and cytokines [4].

### 4.1. Neutrophil Extravasation Cascade

The neutrophil extravasation cascade involves many following steps: capturing, tethering, rolling, adhesion, arrest onto the endothelium, crawling, and diapedesis [74,75]. Once firmly attached, neutrophils flatten and polarize and then crawl onto the endothelial surface in search of a permissive site of extravasation [74]. Neutrophils are captured on activated endothelium at high shear stress made by the blood flow [75,76]. The hydrodynamic drag within venules pushes the neutrophil forward to initiate rolling. The rolling is caused by a quick formation and disconnection of P-selectin glycoprotein ligand-1 (PSGL-1) bonds at the center and rear of the neutrophil and stabilized by long, thin tethers that strengthen adhesive interactions with the EC [75,76]. Neutrophils release oncostatin M, which connects with endothelial gp130 to induce accumulation of P-selectin [77]. Recently, in myosin 1e-deficient mice, “intermittent rolling” was shown, in which rolling neutrophils detach, jump and reattach to continue rolling. Moreover, myosin 1e deficiency contributed to decreased firm adhesion, aberrant crawling and reduced transmigration [78]. The next step is slow rolling, in which the connection between lymphocyte function-associated molecule-1 (LFA-1) on neutrophils and intercellular adhesion molecule 1 (ICAM-1) on the endothelium is crucial [79]. The presence of kindlin-3 and talin-1 and high-affinity conformation of LFA-1 lead to firm adhesion [80]. Actin-binding proteins (ABP), such as Src-kinase associated phosphoprotein 2 (Skap2), myosin 1e and hematopoietic cell-specific lyn substrate (HS1), are critical regulators of slow rolling neutrophils [78,81,82]. HS1 deficiency reduces ICAM-1 binding [81]. Skap2 is necessary for β2-integrin activation and talin-1 and kindlin-3 recruitment, but myosin 1e is required for steady rolling [78,82]. Neutrophil arrest occurs, in most cases, in response to chemokines such as CXCL1 and CXCL2. These chemokines are recognized by G-protein-coupled receptors (GPCR), which are crucial for the acquisition of high-affinity conformation of LFA-1 [83,84,85]. In addition, CXCL1 controls intraluminal and abluminal crawling [83]. Both Skap2 and myosin-1e are also required for chemokine-induced arrest [75]. Then, neutrophils adopt a polarized shape with a cell front enriched with polymerized F-action protrusions and a rear end enriched with actomyosin contractile filament [74].

It was shown that crawling depends on macrophage-1 antigen (Mac-1) because it is diminished when Mac-1 is absent [75,81]. Other studies show that LFA-1 also participates in this stage [86,87]. Moreover, TNF-α induces ICAM-1 up-regulation on lymphatic endothelium, allowing neutrophils to crawl via Mac1/ICAM-1 interactions [88]. The abovementioned myosin 1e depletion disturbs crawling by reducing the number of crawling cells but it does not influence velocity. This suggests that myosin 1e influences the transition from arrest to crawling, but it does not have an influence on crawling itself [75]. The depletion of another factor—myosin heavy chain 9 (MyH9)—significantly reduces neutrophils’ crawling distance and velocity [89]. The following step is diapedesis, which can occur at EC junctions (paracellular migration) or through the EC body (transcellular migration) [74]. Which route is favored depends on stress fibers. When the EC content in stress fibers is high, the paracellular migration is favored because this helps to open EC junctions [74]. In contrast, in transcellular migration, EC content in stress fibers is low, and the EC junctions are tight [74]. Other factors, such as the level of ICAM-1 expression on the EC surface, influence the choice of the way of migration. Transcellular migration was favored in the presence of high ICAM-1 expression on ECs in an in vitro study [90]. Similarly, high concentrations of caveolin-1 in ECs favor the transcellular way, but its low concentration promotes the paracellular way [91]. Transcellular migration could be a compensatory mechanism and a highly regulated process that is favored by heightened adhesion and reduced crawling [74].

The study shows that CXCL1, which is produced mainly by TNF-stimulated ECs and pericytes, supports luminal and sub-EC neutrophil crawling. Conversely, neutrophils are the main producers of CXCL2, and this chemokine is critical for the correct breaching of EC junctions [83]. Some studies reported that neutrophils mostly use transcellular migration in vitro when the EC junctions are too tight, e.g., such as those observed in ECs of the brain [92].

Neutrophils affect the function of ECs by their secretory activity. NET, which is released by neutrophils, is composed of antimicrobial factors like NE and MPO [93]. NE is a serine protease, whereas MPO catalyzes oxidation. Both support killing bacteria and fighting with pathogens [94]. Moreover, the inhibition of neutrophil elastase does not inhibit the cytotoxic effect of NET, but the inhibition of myeloperoxidase reduces NET-induced cytotoxicity [95]. Additionally, neutrophil-derived EVs have a dual influence on endothelial permeability depending on their cargo characteristics [96].

Factors like cathepsin G, MPO, and S100A8 may destroy junction cohesion and escalate permeability. These factors are named barrier-disrupting cargo [97]. Conversely, another factor like annexin 1, which maintains junction integrity and reduces permeability, is called barrier-protecting cargo [98]. A negative influence on endothelial permeability was shown in a study conducted by Ajikumar et al. In this research, neutrophil-derived microvesicles (NMVs), produced by activated neutrophils and internalized by human cerebral microvascular endothelial cells, significantly induce the dysregulation of genes associated with TJs and vesicular transport. NMVs may play a crucial role in modulating the permeability of the BBB during an infection [99].

Endothelial cells produce nitric oxide, which attenuates neutrophil adhesion to the endothelium under whole-blood arterial flow conditions [100]. After stimulation with TNF-α, IL-1β or lipopolysaccharide (LPS), endothelial cells produce a neutrophil chemotactic factor, which influences neutrophil-mediated inflammation [101].

### 4.2. In the Context of Multiple Sclerosis and Ischemic Stroke

MMPs derived from neutrophils contribute to vascular damage. Their higher levels were observed in the CNS during MS and stroke [102,103]. MMP-9 -/- mice, which were used in studies, indicate the critical role of this enzyme in the induction of BBB breakdown in stroke and the EAE model [30,31,104]. MPO, which damages the endothelial cells, is mostly produced by neutrophils and its expression occurs in areas of myeloid cell infiltration in MS and stroke [30,105]. Additionally, the inhibition of MPO in EAE and stroke mice models decreased BBB permeability and diminished disease severity [40,41].

### 4.3. In the Context of Multiple Sclerosis

Neutrophils are one of the first cells to migrate to the CNS in early stages and in relapse phases of MS [46,47]. These cells induce the maturation of pro-IL-1β by the production of proteases and MMPs [106]. IL-1β in the EAE model acts on Th cells to promote pathogenicity in EAE [47]. But not only Th cells are involved in this process. IL-1β is also involved, as it influences other cells, like astrocytes and CNS endothelial cells, and, as a consequence, contributes to BBB disruption and leukocyte recruitment [107]. In the EAE model, migration of neutrophils across the CNS endothelium also triggers the production of IL-1β, which activates receptors for IL-1 on ECs [108,109]. This activation leads to the secretion of proinflammatory factors and results in exacerbated EAE [109]. Additionally, the secretion of IL-1β by neutrophils induces the production of the granulocyte macrophage colony-stimulating factor (GM-CSF) by ECs. This process increases the neuroinflammatory reaction in EAE [108] (Figure 1).

### 4.4. In the Context of Ischemic Stroke

Neutrophils adhere to cerebral microvessels one hour after ischemia in stroke, which indicates that neutrophils are associated with vascular dysfunction at an early stage of stroke [110]. Endothelial β2-integrin ligand ICAM-1 has an impact on neutrophil recruitment in stroke through the BBB. Studies show that ICAM-1 deficiency in mice models ameliorates stroke [111,112]. LCN-2 expression is enhanced during several neuroinflammatory diseases like MS and stroke [113,114]. After ischemic stroke, increased levels of LCN-2 in the blood maintain BBB integrity and reduce damage to endothelial junction proteins [30] (Figure 1).

## 5. Neutrophils and Astrocytes

Astrocytes are the most abundant cells in the CNS and play a decisive role in maintaining the barrier function of the brain microcapillary ECs [115]. Astrocytes contribute to morphological changes in neutrophils, such as a substantial increase in the number of granular particles [116]. Astrocytes enhance the phagocytic capability of neutrophils, decrease neutrophil ROS production, have an inhibitor effect on neutrophils degranulation, increase neutrophil production of pro-inflammatory cytokines like CXCL2, IL-1β, IL-6, and TNF-α, and induce the up-regulation of Akt, phosphorylated Akt, phosphorylated Erk1/2, and phosphorylated p38 levels in neutrophils [116]. Microglia and astrocytes produce IL-1β, which is a firm inducer of endothelial ICAM-1 and VCAM-1 and stimulator of neutrophil infiltration during neuroinflammation [30]. It has been described that reactive astrocytes have two different types in response to different insults: A1 and A2. In an LPS-induced neuroinflammation milieu, A1 astrocytes are activated [117]. A1 astrocytes have proinflammatory and neurotoxic phenotype and secrete IL-1α, TNF-α, and complement component 1q [117]. Moreover, A1 astrocytes promote neutrophil infiltration by upregulating the expression of chemokines (CXCL1, CXCL2, CXCL5) and pro-inflammatory factors (IL-1α, TNF-α) and accelerating BBB breakdown via the release of MMP-9, ROS, and LCN-2 [42]. Additionally, astrocytes contact neutrophils directly during migration. Direct interaction between neutrophils and astrocytes increases the inflammatory response of neutrophils and promotes the expression of their pro-inflammatory factors (IL-1β, IL-6, TNF-α), as well as extending the lifespan of neutrophils and increasing their phagocytic ability [44,116]. Conversely, A2 astrocytes have neuroprotective phenotypes, which can be noticed in ischemia [118]. A2 astrocytes secrete anti-inflammatory and growth factors, like insulin-like growth factor 1 (IGF-1), brain-derived neurotrophic factor (BDNF), and hemopoietin [118].

Neutrophils produce proinflammatory cytokines, like IL-1β, which impact astrocytes and increase their VEGF-A production. In the CNS, VEGF-A disrupts endothelium claudins and occludin, which play crucial roles in junction formation and junction regulation at the BBB, which leads to BBB disruption and leukocytes infiltration to the CNS [12]. Neutrophils also contribute to the maturation and migration of astrocytes by releasing complement components (C1q and C3a) [119].

### 5.1. In the Context of Multiple Sclerosis

Neutrophils during migration across the BBB and activated endothelium produce enhanced amounts of ROS, which, in EAE and MS, damage astrocytes and axons [120]. The migration of neutrophils into the CNS in EAE depends on CXCL1, CXCL2, and CXCL6. In the EAE model, astrocytes with induced CXCL1 expression lead to enhanced EAE severity and increase the amount of migrated neutrophils to the brain and spinal cord [121]. MMPs secreted by neutrophils can cleave dystroglycan, which attaches the astrocytic endfeet to the BBB. Researchers found that mice with double MMP-2 and MMP-9 knockout were resistant to EAE due to the limited cleavage of dystroglycan and inhibited leukocyte migration to the CNS [31]. Neutrophils, by the secretion of IL-1β, activate NLRP3 inflammasome. The activation of this inflammasome in glial cells (astrocytes) increases neuroinflammation and exacerbates EAE [122,123] (Figure 2).

### 5.2. In the Context of Ischemic Stroke

A2 astrocytes limit the propagation of neutrophils in the area around the infarction by forming glial scars and phagocytosis of neutrophils [42]. Similarly to neutrophils, A1 astrocytes play a crucial role in early stages after ischemic stroke [124]. A2 astrocytes are crucial in a later stage, as in the brain, they gradually increase 72 h after ischemic stroke [125]. Damaged neutrophils release DAMPs after ischemic stroke, which activates astrocytes. In special circumstances like a disease or injury, astrocytes respond by reactive astrogliosis, which exhibits increased expression of ROS, LCN-2, VEGF, cytokines like IL-1 and IL-6, and MMP-9, which, in turn, are involved in BBB breakdown [42,43,126]. Astrocyte secretion of VEGF is increased in the penumbra after ischemia [42]. VEGF induces BBB breakdown by downregulating the TJs proteins of ECs directly and also by activating MMP-9 to degrade connexins [12,127]. On the other hand, astrocytes may have a neuroprotective role and produce anti-inflammatory factors in ischemia [118]. Neutrophils can support the anti-inflammatory role of astrocytes by the secretion of high-mobility group box 1 (HMGB1), which is involved in the stabilization of nucleosomes and DNA repair. HMGB1 contributes to astrocyte and neurovascular repair in the late stage of ischemic injury [128] (Figure 2).

## 6. Neutrophils and Pericytes

PCs are located between endothelial cells and astrocyte end-feet [9]. These cells have the ability to contract, which allows them to regulate cerebral blood flow by controlling the luminal diameter [9]. PCs can construct tunnelling nanotubes that regulate neurovascular coupling and control capillary blood flow [129]. ECs connected with PCs are more resistant to apoptosis than isolated ECs. This connection supports the role of PCs in supporting the structural integrity of the BBB [130]. Pericytes are also crucial for the induction of barrier characteristics in the endothelial cells including formation of TJs [2]. Moreover, PCs participate in the regulation of ECs differentiation, proliferation and vesicle trafficking in the CNS [19,131].

Pericytes participate in neutrophil transmigration through venular walls. An in vivo study showed that after EC migration, neutrophils crawl along pericyte processes to gaps between adjacent pericytes [48]. This step is dependent on ICAM-1, Mac-1, and LFA-1 [48]. Proinflammatory cytokines like TNF and IL-1β can change the shape of pericytes [48]. Pericytes increase the cell length and reduce the cell width, which leads to enlarged gaps between adjacent cells in venules. These gaps are used as exit points by neutrophils breaching hereby the venular wall [48,132]. Pericytes can produce mediators (e.g., CXCL8), which increase neutrophil transmigration through the endothelial BBB model [49]. Moreover, pericyte-mediated neutrophil transmigration is independent of endothelial gene expression [49]. Another study showed that after stimulation with proinflammatory factors like LPS, TNF-α, or IL-1β, pericytes increase neutrophil transmigration by IL-8 in an in vitro model [48]. This effect was remarkably decreased by inhibitors of MMP-2 and MMP-9, because this inhibition increases the adhesion of neutrophils to pericytes and thus suppresses the transmigration of neutrophils [48]. According to Regis et al., final steps of neutrophil diapedesis are regulated by perivascular mast cells (MCs)-IL-17A-pericyte axis [133]. MCs secreted IL-17A upon inflammation, and this IL-17A led to ICAM-1 and CXCL1 enhanced expression in pericytes, which promoted efficient diapedesis of circulating neutrophils in vivo [133]. They also showed that MC deficiency and blockade of IL-17A lead to impaired neutrophil transmigration and breach of the pericyte layer [133]. Another study showed that proinflammatory cytokine IL-17, alone or with TNF, significantly changed inflammatory gene expression in PCs but not in ECs. IL-17-activated PCs can modify neutrophil functions [134]. After activation, neutrophil phagocytic capacity was enhanced, and the production of proinflammatory molecules (e.g., TNF, IL-1α, IL-1β, and IL-8) increased [134]. Moreover, IL-17-activated PCs can prolong neutrophil survival by secreting GM-CSF, G-CSF and caspase-9 activation [134]. The secretion of IL-1β neutrophils increases microvascular permeability in pericyte/endothelial cell co-cultures [135].

### 6.1. In the Context of Multiple Sclerosis

Pericytes can suppress the expression of activated leukocyte cell adhesion molecule (ALCAM), which, in patients with MS, is upregulated in CNS vessels and increases leukocyte migration through endothelial cells [136]. In a study on EAE, PCs stimulated with Chondroitin sulfate proteoglycans, a family of extracellular matrix protein, increased macrophage migration to the CNS, which in turn increased neuroinflammation in EAE [137]. On the other hand, PCs may participate in remyelination, and loss of PCs delays remyelination in in vivo study [138]. PCs secrete proteins such as A-kinase anchor protein 12 (AKAP12) and laminin subunit alpha 2 (lama 2), which can mediate remyelination [139,140]. AKAP12 could promote oligodendrocyte progenitor cell maturation, and AKAP12-deficient mice have less myelin in comparison to control mice [139]. Moreover, lama 2 may stimulate oligodendrocyte progenitor cell differentiation [138]. Neutrophils can activate NLRP3 by releasing IL-1β. Activation of NLRP3 inflammasome was found in CNS cells, including pericytes. In MS, protein levels of NLRP3 inflammasome components are increased [122,123] (Figure 3).

### 6.2. In the Context of Ischemic Stroke

After ischemic stroke, pericytes detach from the basement membrane, contract, and undergo cell death. The result of this process is increased BBB permeability [42,44]. In ischemia, PCs are able to secrete MMP-9, which can damage the capillary wall and facilitate migration of leukocytes through the BBB [45]. PC can also act as immune cells and secrete proinflammatory factors like ROS, MMPs and participate in increasing BBB permeability after stroke [50]. On the other side, pericyte constriction could be helpful by shutting down damaged microvessels, suppressing BBB disruption, and preventing edema after stroke [44].

After stroke, neutrophils release NETs, which are digested by DNase 1. This step can support PCs by increasing PC coverage on micro-vessels and forming new vessels. In consequence, this process leads to stroke recovery [141] (Figure 3).

## 7. New Therapeutic Strategies for MS and IS

Nowadays, there are no effective treatments for MS and IS. Treatments that are used in patients with MS are disease-modifying therapies (DMTs), which modulate or inhibit the immune system, suppress inflammation in the CNS, and reduce the relapse rate and delay onset of disability [142]. Additionally, DMTs have many side effects, starting from flu-like syndrome and headache and ending with cardiac arrhythmias and malignancy [143]. Patients with acute phase of IS can receive thrombolytic and endovascular therapies. Despite recent advances, these therapies can still contribute to patient mortality or disability. Additionally, secondary prevention is restricted by a limited choice of treatment and non-adherence to doctor’s recommendation by patients [144].

The treatment, which we can offer to patients with MS and IS, is not as effective as it should be. Hence, cell-based therapies have been recently gaining more attention. These therapies are classified into immunoablation, followed by autologous hematopoietic stem cell transplantation (I/AHSCT), transplantation of mesenchymal stem cells (MSCs), other stem cells, and oligodendrocyte progenitor cells (OPCs) or OPC-like inducible pluripotent stem cells (iPSCs) [145].

Many studies on MS imply that I/AHSCT has relevant efficacy in the inhibition of inflammatory reaction, resulting in significant improvements in the EDSS score and relapse incidence. Additionally, visual acuity and visual evoked response latency appeared to improve as well [145,146,147,148,149,150]. Moreover, MSCs and bone marrow-derived cells have potential therapeutic value in MS. Studies reveal that these therapies caused remyelination, reduced gliotic scar formation, angiogenesis, suppressed inflammation, and resulted in immune modulation and neuroprotection [145,151,152,153,154,155]. Moreover, exosomes derived from mesenchymal stem cells (MSC-Exos) decrease Th1 and Th17 expression and lead to myelin regeneration [156]. Therapies utilizing OPC or iPSC transplantation can potentially result in the replacement of damaged or lost myelin-generating oligodendrocytes [145].

In ischemic stroke, cell-based therapy is focused on neurogenesis and angio- and arteriogenesis by the suppression of inflammatory cytokines like IL-6, IL-2, TNF-α, and IFN-γ and by the suppression of macrophage activation and endothelial injury [157].

Lai et al. found that MSC-Exos reduce the neurotoxic effects of astrocytes by inhibiting the expression of Traf6 and Irak1, which are involved in NFκB pathway activation of neuroinflammatory response [158]. Another study reveals that MSC-derived EVs diminish NLRP-3-inflammasome, decrease secretion of pro-inflammatory cytokines, and reduce astrogliosis in mouse astrocytes [159].

Cell-based therapies for MS, IS, and other diseases are experimental, and at present, there is no clear evidence that they are effective. These methods should be assessed in rigorous clinical trials [145].

## 8. Therapeutic Neutrophil Function

Neutrophils have the ability to secrete proinflammatory factors like ROS, NET, MPO, MMPs, which influence BBB disruption and also contribute to development of neuroinflammation, which is the cause of CNS diseases like MS and IS [30]. However, neutrophils are not a homogeneous population. The TAN classification distinguishes N1 and N2 neutrophils, which have pro-inflammatory and anti-inflammatory features, respectively [66,67]. In ischemic stroke, N1 neutrophils increase neurovascular damage in the acute stage. In contrast, N2 neutrophils serve as neuroprotectors in later stages [67,160]. In MS, N1 neutrophils play a greater role, but studies using single-cell RNA sequencing technology reveal other clusters of neutrophils, which can act in different ways [161,162]. Additionally, neutrophils can produce EVs, which can easily cross the BBB and also exhibit pro- or anti-inflammatory features depending on neutrophils and the kind of stimuli [64,65]. Based on these studies, neutrophils can exacerbate, but also alleviate, CNS diseases, depending on the population, milieu and stage of the disease. This suggests that neutrophils are a good option in the search for potential treatments against CNS diseases.

Neutrophil-derived nanovesicles (NNVs) can represent a promising strategy for MS and IS treatment. Researchers developed NNVs to enhance the efficiency of myelin debris clearance by microglia for MS therapy. NNV treatment upregulates the expression of nuclear factor E2-related factor 2 (Nrf2) in microglia, which results in their enhanced phagocytic activity [163]. Moreover, NNVs can be used for the delivery of therapeutics to ischemic stroke lesions. In an in vivo study, scientists report a drug delivery system, composed of neutrophil membrane-derived nanovesicles, loaded with Resolvin D2, which decreases inflammation [164]. They observed that these nanovesicles significantly alleviated inflammation in ischemic stroke and improved mouse neurological functions [164]. These studies give us a new approach to treatment using neutrophils, which, in the future, can be used as therapeutic techniques to cure CNS diseases.

## 9. Discussion

The functions of the BBB are to protect the CNS from toxic factors and provide communication between the periphery and the CNS [1]. These functions can be fulfilled because the BBB is a selective semi-permeable membrane, which stays relatively impermeable under proper conditions [1,2]. Pathologic conditions like the effect of hypoxia–ischemia, occurring in ischemic stroke or autoimmune background, which is related to multiple sclerosis, increase BBB permeability [2,38]. The pathogeneses of these diseases are not fully understood, but we know that neutrophils and their secretory factors have influenced both BBB permeability and these diseases [30,57,141,161,165,166,167].

In recent years, neutrophils have been receiving more attention among researchers, mostly because of the development of new techniques, which showed complex neutrophil heterogeneity. Firstly, neutrophils are classified as tumor-associated neutrophils into two groups: N1 phenotype and N2 phenotype [67,68]. N1 neutrophils have pro-inflammatory and N2 neutrophils have anti-inflammatory characteristics [66,67]. In the acute stage of ischemic stroke, N1 neutrophils play a greater role and exacerbate neurovascular damage, whereas N2 neutrophils gain more importance after this stage and support neurovascular repair [42,160]. In MS, scientists do not observe this distinctive feature of neutrophils between relapse and remission. In studies with relapsing-remitting multiple sclerosis (RRMS) and EAE, neutrophil phenotype is similar to the N1 phenotype [162]. Thus, more investigations to determine the impact of different subpopulations of neutrophils on stage or clinical phenotype of MS and IS are needed.

Neutrophils, like most other cells, have the ability to secrete extracellular vesicles, which can cross the BBB and transfer information [64,168]. Neutrophil-derived EVs, which contain cargo, including S100A8, S100A9, MPO, and cathepsin G, increase BBB permeability and damage junction integrity [97]. On the other hand, cargo, such as annexin 1, maintains junction integrity and minimizes permeability [98].

Endothelial cells are one of the cellular elements of the BBB. The relationship between neutrophils and ECs in the context of the neutrophil extravasation cascade is quite well known [74,75,76,78,80,81,82,83,84,85,86,87,88,89,92]. However, this relationship with MS and IS is less well investigated. Research has shown that neutrophils secrete MMPs, which cause vascular damage and MPO, which induces damage to ECs. In MS and IS, higher levels of MMPs and MPO have been observed [30,102,103,105]. MMPs contribute to the maturation of pro-IL-1β, which promotes pathogenicity in EAE [106,107]. IL-1β influences astrocytes and CNS ECs, which cause BBB disruption [107]. Moreover, in stroke, endothelial β2-integrin ligand ICAM-1 influences neutrophil recruitment through the BBB [111,112]. These examples are indicative of a destructive influence of neutrophils on ECs, but neutrophil-derived EVs present both a positive and negative impact on endothelial permeability [96]. The cargo of these EVs can contain factors, which are barrier-disrupting or barrier-protecting [97,98]. Neutrophil-derived EVs highlight a complex relationship between neutrophils and ECs.

Astrocytes are other cellular elements of the BBB. Similarly to neutrophils, astrocytes can be categorized into A1 and A2 subpopulations [117]. A1 astrocytes support neutrophil infiltration by increasing pro-inflammatory factors (IL-1α, TNF-α) and BBB permeability and by secreting MMP-9 and LCN-2 [42]. Direct interaction between neutrophils and astrocytes enhances the inflammatory reaction of neutrophils, promotes the expression of proinflammatory factors (IL-1β and TNF-α), and increases neutrophils’ phagocytic ability [42,116]. On the other hand, A2 astrocytes limit the propagation of neutrophils in the infarct area and their phagocytic capacity [42]. This implies that various astrocyte subpopulations may support neutrophils in their pro- and anti-inflammatory response.

In IS, similarly to neutrophils, A1 astrocytes display a greater role in early stages following IS, and A2 astrocytes play a more important role in later stages following IS [124,125]. This might explain why astrocyte-derived EVs in studies on IS, on the one hand, improve survival in neurons under hypoxia, inhibit autophagy and ameliorate neuronal damage, and, on the other hand, suppress repair and reduce axon growth [169,170,171,172]. Little is known about the effects of the interplay between such astrocytic subtypes with various functional types of neutrophils and their role in various IS phases.

After stroke, PCs can cause increased BBB permeability by facilitating migration of leukocytes through the BBB and by secreting proinflammatory factors [45,50]. However, PCs may also decrease damage to microvessels and suppress BBB disruption [44]. After stroke, neutrophils support PCs in forming new vessels, which leads to improved stroke recovery [141].

In this article, we presented many examples of the influence of neutrophils on ECs, astrocytes and PCs in common diseases of the CNS, like MS and IS. The impact is both positive and negative. The important issue is how we can enhance positive effects and diminish the negative influence.

## 10. Conclusions 

Neuroinflammation, BBB disruption, the recruitment of neutrophils and other peripheral cells into the CNS, and the interaction between cellular elements of the BBB are the main causes of CNS diseases, such as MS and IS. Recent studies reveal that neutrophils have various subpopulations, which can have pro- or anti-inflammatory properties, depending on the milieu, activity of other cells, and even stages of disease. Astrocytes, being elements of the BBB, are also a heterogenous population. This diversity results in a more complex interaction with neutrophils. Moreover, interactions between pericytes, endothelial cells, astrocytes and neutrophils can have both positive and negative effects. A detailed understanding of the relationship between BBB elements and different neutrophil subtypes in the context of such complex diseases as IS and MS requires additional research, the application of advanced techniques, and standardization of the method characterizing neutrophil subpopulations. In this area, scientists have a lot of possibilities to search for new therapeutic approaches.

## 11. Future Perspectives

The development of advanced techniques, like single-cell RNA sequencing technology, provides the opportunity to create more complicated classification of neutrophils and astrocytes. A study with a mouse IS model presented four distinct neutrophil clusters. Another study showed three clusters of neutrophils [165,166]. Similarly, a study with EAE revealed three clusters of neutrophils [161]. Interestingly, in EAE, as well as in middle cerebral artery occlusion (MCAO), other clusters are more visible in acute stages rather than in chronic stages [161,165,166]. In a study using single-nucleus RNA sequencing, researchers also revealed “astrocytes inflamed in MS”, which presented neurodegenerative programming [173]. In another study, scientists indicate that exosomes derived from astrocytes of MS patients can modify the activity of human T cells [174]. The identification of the role of such astrocyte subtypes or their EVs in the communication with neutrophils in the MS course requires additional studies.

Due to recent studies, we have gained more data about various subpopulations of neutrophils, but this information is incomplete. There are different types of neutrophil classifications, which are based on the expression of neutrophil receptors, secretory activity, or sequencing neutrophil RNA. The variety of available techniques may cause difficulty in creating a standardized classification of subpopulations of these cells. Hereby, we still have little information about the complexity of different subpopulations of neutrophils and their interaction with cellular components of the BBB, i.e., astrocytes, pericytes, and endothelial cells. There is still wide scope for future investigation. Knowledge about mechanisms and reactions between these elements can contribute in the future to developing new methods of treatment or earlier detection of common diseases of the CNS like MS or IS.

## Figures and Tables

**Figure 1 ijms-26-04437-f001:**
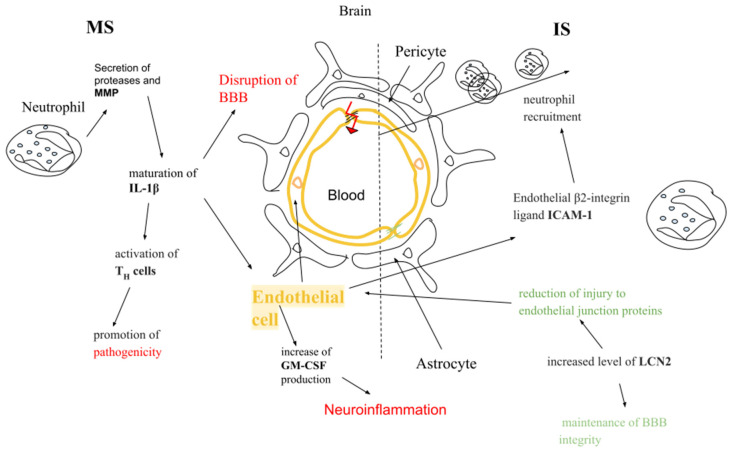
Comparison of interaction between neutrophils and endothelial cells in the pathogenesis of multiple sclerosis (MS) and ischemic stroke (IS). MMP—matrix metalloproteinases, BBB—blood–brain barrier, GM-CSF—granulocyte macrophage colony-stimulating factor, ICAM-1—intercellular adhesion molecule 1, LCN-2—lipocalin 2. Green text—protective effect, red text—detrimental effect.

**Figure 2 ijms-26-04437-f002:**
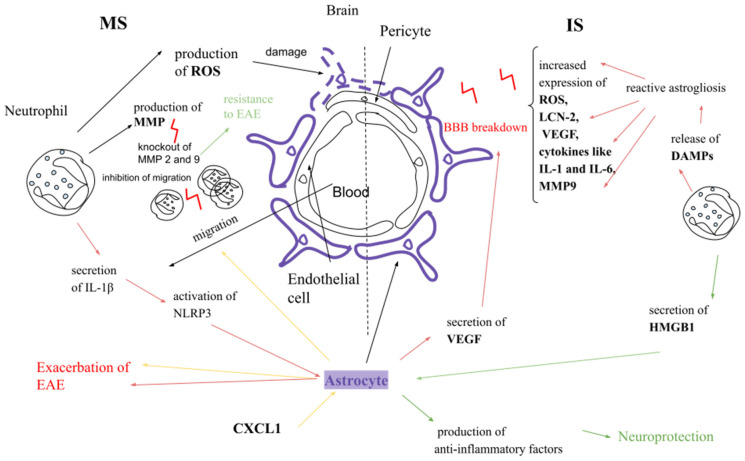
Comparison of interaction between neutrophils and astrocytes in the pathogenesis of MS and IS. ROS—reactive oxygen species, EAE—experimental autoimmune encephalomyelitis, NLRP3 -Nod-like receptor containing pyrin domain 3, VEGF—vascular endothelial growth factor, DAMPs—damage-associated molecular patterns, HMGB1—high-mobility group box 1. Green lines—protective effect, red lines—detrimental effect, yellow lines—CXCL1 effect.

**Figure 3 ijms-26-04437-f003:**
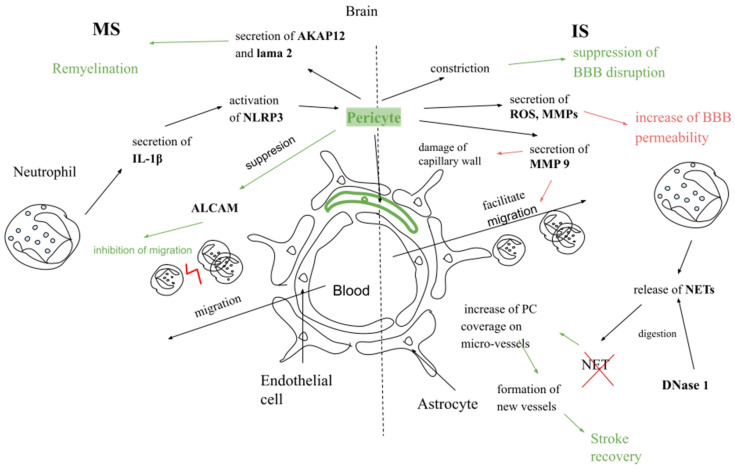
Comparison of interaction between neutrophils and pericytes (PC) in the pathogenesis of MS and IS. AKAP12—A-kinase anchor protein 12, lama 2—laminin subunit alpha 2, ALCAM—activated leukocyte cell adhesion molecule, NETs—neutrophil extracellular traps, DNase 1—deoxyribonuclease I. Green lines—protective effect, red lines—detrimental effect.

## Data Availability

The data presented in this study are available on request from the corresponding author. The data are not publicly available due to privacy.

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
