# Peer review of "Interaction Between Neutrophils and Elements of the Blood–Brain Barrier in the Context of Multiple Sclerosis and Ischemic Stroke"

_ijms, 2025, doi:10.3390/ijms26094437_

Round 1
Reviewer 1 Report
Comments and Suggestions for Authors
The paper provides important information in the field of study. The manuscript structure is well organized, and the evidence demonstrated is adequate. However, there are a few concerns that need to be addressed.
It would be convenient that the authors describe more demyelinating diseases, according to the title "The Interaction Between Neutrophils and Elements of the Blood-Brain Barrier in the Context of Demyelination and Ischemia", not only "but we have chosen these two brain diseases to discuss two various pathomechanisms. Firstly, multiple sclerosis as autoimmune disease and secondly ischemic stroke as vascular disease with hypoxia-ischemia mechanism"
The paper would be enriched by adding a section on possible treatments that regulate the interaction between neutrophils and elements of the blood-brain barrier.
Author Response
Comments 1: The paper provides important information in the field of study. The manuscript structure is well organized, and the evidence demonstrated is adequate. However, there are a few concerns that need to be addressed.
It would be convenient that the authors describe more demyelinating diseases, according to the title "The Interaction Between Neutrophils and Elements of the Blood-Brain Barrier in the Context of Demyelination and Ischemia", not only "but we have chosen these two brain diseases to discuss two various pathomechanisms. Firstly, multiple sclerosis as autoimmune disease and secondly ischemic stroke as vascular disease with hypoxia-ischemia mechanism"
Response 1: Thank you for your suggestion. We agree that analysis of only one disease in context of demyelination is insufficient. However, our paper is already extensive and discusses the complex interaction between neutrophils and cellular elements of BBB based on multiple sclerosis- and ischemic stroke-related results, so we decided to change title :”The Interaction Between Neutrophils and Elements of the Blood-Brain Barrier in the Context of Multiple Sclerosis and Ischemic Stroke”.
Comments 2: The paper would be enriched by adding a section on possible treatments that regulate the interaction between neutrophils and elements of the blood-brain barrier.
Response 2: According to your suggestions we have described the current knowledge about possible treatments in two sections: “New therapeutic strategies for MS and IS” and “Therapeutic neutrophil function”.
Reviewer 2 Report
Comments and Suggestions for Authors
The authors of the manuscript "The Interaction Between Neutrophils and Elements of the Blood-Brain Barrier in the Context of Demyelination and Ischemia" provide a summary of both foundational knowledge and some recent insights related to the topic. However, the manuscript would benefit from more scientifically structured language, as many sentences are overly simplistic in their current form.
Additionally, several typographical and technical errors were noted throughout the manuscript, and a thorough English language edit is strongly recommended. All abbreviations should be clearly defined upon first use to ensure clarity for the reader.
Importantly, the Conclusion and Future Perspectives sections are missing and should be included to complete the manuscript.
Comments on the Quality of English LanguageEnglish language editing is highly recommended.
Author Response
Comments 1: The authors of the manuscript "The Interaction Between Neutrophils and Elements of the Blood-Brain Barrier in the Context of Demyelination and Ischemia" provide a summary of both foundational knowledge and some recent insights related to the topic. However, the manuscript would benefit from more scientifically structured language, as many sentences are overly simplistic in their current form.
Response 1: Thank you for your suggestion. The article was checked and corrected to improve scientific language.
Comments 2: Additionally, several typographical and technical errors were noted throughout the manuscript, and a thorough English language edit is strongly recommended. All abbreviations should be clearly defined upon first use to ensure clarity for the reader.
Response 2: The article was checked and corrected to improve English language and deleted typographical and technical errors. Moreover, abbreviations were corrected.
Comments 3: Importantly, the Conclusion and Future Perspectives sections are missing and should be included to complete the manuscript.
Response 3: According to your suggestion, we have added “Conclusion” and “Future Perspectives” sections.
Reviewer 3 Report
Comments and Suggestions for Authors
In this review the authors focused on the interaction between neutrophils and elements of the blood-brain barrier in pathologies such as multiples sclerosis and ischemia. Manuscript is well-written, but I have some suggestions to improve the manuscript.
1) MS and IS share neuroinflammation, a pathological mechanism leading to the glial cell activation, disruption of the BBB and recruitment of peripheral immune cells in the CNS. Although the importance of neuroinflammation in both pathologies, this aspect was poorly described in the manuscript. I suggest to add a paragraph dedicated to the role of neuroinflammation in these pathologies and the effects of neuroinflammation on the components of the blood brain barrier.
2) For a greater completeness, I suggest to add a paragraph to describe the new therapeutical strategies for MS and IS that can target the BBB components and neutrophils. These therapies have the aim to reduce neuroinflammation, neutrophil infiltration and improve the tightness of BBB. Stem-cell therapies and cell reprogramming therapies are also appreciated.
Author Response
Comments 1: In this review the authors focused on the interaction between neutrophils and elements of the blood-brain barrier in pathologies such as multiples sclerosis and ischemia. Manuscript is well-written, but I have some suggestions to improve the manuscript.
1) MS and IS share neuroinflammation, a pathological mechanism leading to the glial cell activation, disruption of the BBB and recruitment of peripheral immune cells in the CNS. Although the importance of neuroinflammation in both pathologies, this aspect was poorly described in the manuscript. I suggest to add a paragraph dedicated to the role of neuroinflammation in these pathologies and the effects of neuroinflammation on the components of the blood brain barrier.
Response 1: Thank you for your suggestion, we added “Role of neuroinflammation” section, in which we described role of neuroinflammation in pathogenesis of multiple sclerosis and ischemic stroke and also effects of neuroinflammation in the context of the components of the blood brain barrier.
Comments 2: 2) For a greater completeness, I suggest to add a paragraph to describe the new therapeutical strategies for MS and IS that can target the BBB components and neutrophils. These therapies have the aim to reduce neuroinflammation, neutrophil infiltration and improve the tightness of BBB. Stem-cell therapies and cell reprogramming therapies are also appreciated.
Response 2: According to your suggestion, we added “New therapeutic strategies for MS and IS” section.
Reviewer 4 Report
Comments and Suggestions for Authors
The authors have submitted a review article of illustrating a current knowledge regarding impact of neutrophils on the cellular components of the blood-brain barrier (BBB) such as endothelial cells, astrocytes, and pericytes. BBB dysfunction is associated with changes in interaction between components of BBB in some human diseases such as multiple sclerosis and stroke. It is of important to understand how pathogenesis of such diseases under association with BBB dysfunction after the interaction with neutrophils. The authors searched a range of eligible literature, from well-known classical, and latest research regarding an association of the BBB components with neutrophils which appears to be a key mechanism of BBB dysfunction (for instance, BBB permeability) which causes CNS diseases. The discovery of the commonality that changes in BBB function are closely related to the onset of various diseases provides a new perspective for the search for treatments for these diseases for which effective treatments or prevention methods have not yet been established. This issue is of interest, and impact of their review is strong. My overall concern with the review describing the current available data regarding neutrophil associated BBB dysfunction which might cause CNS diseases listed in this review is that information provided may offer something substantial that helps advance our understanding of effective management which draws novel class of effective anti-CNS diseases available in clinic. The reference list may be useful for readers who are interested in this issue.
To strengthen authors’ perspectives, the authors are strongly recommended to add a “therapeutic neutrophil function” prediction, for instance. When neutrophils play a role in central nervous system diseases, attention must be paid to whether they are the cause of the disease or whether they work to alleviate the induced disease.
Author Response
Comments 1: The authors have submitted a review article of illustrating a current knowledge regarding impact of neutrophils on the cellular components of the blood-brain barrier (BBB) such as endothelial cells, astrocytes, and pericytes. BBB dysfunction is associated with changes in interaction between components of BBB in some human diseases such as multiple sclerosis and stroke. It is of important to understand how pathogenesis of such diseases under association with BBB dysfunction after the interaction with neutrophils. The authors searched a range of eligible literature, from well-known classical, and latest research regarding an association of the BBB components with neutrophils which appears to be a key mechanism of BBB dysfunction (for instance, BBB permeability) which causes CNS diseases. The discovery of the commonality that changes in BBB function are closely related to the onset of various diseases provides a new perspective for the search for treatments for these diseases for which effective treatments or prevention methods have not yet been established. This issue is of interest, and impact of their review is strong. My overall concern with the review describing the current available data regarding neutrophil associated BBB dysfunction which might cause CNS diseases listed in this review is that information provided may offer something substantial that helps advance our understanding of effective management which draws novel class of effective anti-CNS diseases available in clinic. The reference list may be useful for readers who are interested in this issue.
Response 1: Thank you for your kind opinion, we appreciate that topic, which is noteworthy for us and is also important and useful for other researchers.To emphasize how knowledge about neutrophils and cellular elements of the blood brain barrier influence MS and IS, we added two sections: “New therapeutic strategies for MS and IS” and “Therapeutic neutrophil function”.
Comments 2: To strengthen authors’ perspectives, the authors are strongly recommended to add a “therapeutic neutrophil function” prediction, for instance. When neutrophils play a role in central nervous system diseases, attention must be paid to whether they are the cause of the disease or whether they work to alleviate the induced disease.
Response 2: Thank you for your suggestion, we added the “Therapeutic neutrophil function” section.
Round 2
Reviewer 3 Report
Comments and Suggestions for Authors
I ask the authors to make some adjustments.
1) Therapeutical strategies described in the article are not cited in the abstract. Please make reference to the therapeutical strategies in the abstract.
2) Line 91: please provide a short description about the impact of neuroinflammation on BBB dysfunction.
3) I suggest the authors to denominate paragraph 6 "Neuroinflammation-induced BBB dysfunction in MS and IS". In addition, please move the paragraph after the introduction.
Author Response
Comments 1. Therapeutical strategies described in the article are not cited in the abstract. Please make reference to the therapeutical strategies in the abstract.
Response 1: Thank you for your suggestion, we added reference to the therapeutical strategies in the abstract.
Comments 2. Line 91: please provide a short description about the impact of neuroinflammation on BBB dysfunction.
Response 2: We inserted a brief description about the impact of neuroinflammation on BBB dysfunction at the end of paragraph about function of BBB in the introduction, because line 91 in our version of manuscript is in part about MS.
Comments 3. I suggest the authors to denominate paragraph 6 "Neuroinflammation-induced BBB dysfunction in MS and IS". In addition, please move the paragraph after the introduction.
Response 3: According to your suggestion, we denominated paragraph 6 and moved this paragraph after introduction.
Reviewer 4 Report
Comments and Suggestions for Authors
The authors have adequately addressed concerns with the previous version of the manuscript. I have no more comments, and recommend that this manuscript is acceptable for publication in the journal IJMS.
Author Response
Comments 1: The authors have adequately addressed concerns with the previous version of the manuscript. I have no more comments, and recommend that this manuscript is acceptable for publication in the journal IJMS.
Response 1: Thank you for your comment.